# Study protocol for a comparative effectiveness evaluation of abiraterone acetate against enzalutamide: a longitudinal study based on Swedish administrative registers

Per Johansson ,[1,2] Paulina Jonéus,[3] Sophie Langenskiöld [4]

[1]Statistics, Uppsala University, Uppsala, Sweden
[2]YMSC, Tsinghua University, Beijing, China
[3]Department of Statistics, Uppsala University, Sweden
[4]Department of Medical Sciences, Uppsala University, Uppsala, Sweden

**Correspondence to**
Dr Sophie Langenskiöld;
sophie.langenskiold@medsci.uu.se

## ABSTRACT

**Introduction** This paper presents a study protocol for a comparative effectiveness evaluation of abiraterone acetate against enzalutamide in clinical practice, two cancer drugs given to patients suffering from advanced prostate cancer.

**Method and analysis** The protocol designs a comparative-effectiveness analysis of abiraterone acetate against enzalutamide. With the substantial number of covariates a two-step procedure is suggested in choosing relevant covariates in the matching design. In the first step, an exploratory factor analysis reduces the dimension of a large set of continuous covariates to nine factors. In the second step, we reduce the dimension of the covariates, interactions and second order terms for the continuous covariates using propensity score estimation. The final design makes use of a genetic matching algorithm. The study protocol provides a detailed statistical analysis plan of the analysis sample derived from the matching design. The analysis will make use of linear regression and robust inference adjusted for multisignificance testing.

**Discussion** As in a randomised experiment the focus is on the design of the assignment to treatment. This allows the publication of this preanalysis plan before having access to outcome data. This means that the p values will be correct if the maintained assumption of uncounfoundedness is valid. Given that is p-hacking is substantial problem in empirical research, this is a substantial strength of this study. However, while design yields, balance on the observed covariates one cannot discard the possibility that unobserved confounders are not balanced. For that reason, sensitivity tests for the maintained assumption of uncounfoundedness are presented.

**Ethics and dissemination** The study was approved by the Regional Ethical Review Board in Uppsala, Sweden (Dnr 2017/482). Results will be published in a peer-reviewed journal and distributed to relevant stakeholders in healthcare.

## INTRODUCTION

Prostate cancer (PC) is the most commonly diagnosed form of cancer in Sweden. In 2016, for example, 10 473 additional patients were diagnosed with PC creating a pool of 107 752 patients.[1 2] PC is also the second leading cause of cancer death, and almost all mortalities arise when the patients have progressed to the advanced stage, metastatic castrate resistant PC (mCRPC).

Various treatment alternatives are available for patients with mCRPC. In the last two decades, chemotherapy and novel hormone therapy (NHT) medications have revolutionised the treatment in mCRPC patients.[3–8] This paper presents a study protocol for the design of a comparative effectiveness evaluation of two of these NHTs, that is, abiraterone acetate (AA) in combination with prednisone and enzalutamide (ENZ). The evaluation concerns their use in clinical practice from June 2015, which corresponds to the period when these drugs where reimbursed for mCRPC patients.

Data are collected from population registers administrated by the National Board of Health and Welfare (NBHW), and Statistics Sweden (SCB). The population is restricted to all men in the NBHW register with a PC

### Strengths and limitations of this study

► The study uses up-to-date mixed methods to design an observational study for a comparative effectiveness study of abiraterone acetate against enzalutamide of four outcomes.
► The study replicates a randomised study ex-post, by achieving comparable groups on patients' historical health and socioeconomic status based on rich linked population registries.
► As outcome data will be added after the publication of the protocol; the design and analysis are consequently not affected by post-treatment variables.
► Sensitivity analyses to evaluate potential remaining confounding are suggested in the protocol.

diagnose before 2017 as only these patients were expected to progress to mCRPC during the period for which we planned to have outcome data. We restrict the population to patients prescribed AA or ENZ during the period 1 June 2015 to 15 June 2018, that is, the period in which AA and ENZ were only prescribed for mCRPC.

The aim of the design is to estimate the comparative effectiveness of AA against ENZ using detailed data from the population registers and up to date statistical methods combined with an understanding of the prescription pattern of the patients obtained from a qualitative study with twelve urologists and oncologists at different hospitals and across different specialisation all over Sweden. The main outcome of interest is overall mortality. Mortality data will be added after the publication of this study design. In addition, the comparative effectiveness on two secondary outcomes (pain and skeleton related events) will be studied. Treatment length and prostate specific mortality will be added as exploratory endpoints.

## METHOD AND ANALYSIS
### Study population
This nationwide longitudinal study uses patient-level data from multiple population registries administrated by NBHW and SCB. Data from these registries are linked using a unique serial number created by SCB.

The population under study is defined using the cancer register. We first identify the number of men with a PC diagnosis before 2017 and the year of their diagnosis. The annual number of men with diagnoses as a proportion of the men in the population has increased more than 100% between 1968 and 2001, from just above 0.1% in 1986 to more than 0.2% in 2016 (see online supplemental figure S1 in online supplemental appendix). We identify 243 535 unique patients with PC (ICD-10 code C61.9 or earlier codes ICD-7 177 and ICD-9 185.9). If a patient for some reason has been diagnosed multiple times with this diagnosis, the first observed time period is used (1661 patients appear multiple times, 1623 of these diagnosed on two or more different dates and 181 patients are diagnosed at different hospitals).

The population is restricted to all men collecting a prescription of AA or ENZ during the period 1 June 2015 to 15 June 2018. The reasons for the time restriction are: (1) that almost no one was treated with these drugs before the reimbursement of AA and ENZ in June and July 2015, respectively, and (2) that AA was additionally reimbursed in combination with Androgen Deprivation Therapy (ADT) in patients with high-risk castration sensitive PC in June 15 2018.

The restriction leaves us with a total of 4 601 patients in the study population. Some of the patients were prescribed AA or ENZ before the subsidisation but collected the drug after June 2015, these are included in the sample. For this population the year of the diagnosis ranges between 1986 and 2016. Consequently, there is substantial variation in the time to be prescribed AA or

ENZ from the date of diagnosis. This, so called, time to prescription is most likely an important covariate.

About 10% of the patients were prescribed both AA and ENZ over the years. We allocate these patients to the two samples AA and ENZ-takers based on their first prescription of one of the two drugs which means that the results should be interpreted as an intention to treat analysis. The number of AA and ENZ patients is increasing until 2016 (see online supplemental table S1 in online supplemental appendix).

The prescription of the two drugs varies over the 21 county councils, here after denoted counties, the responsible body for healthcare in Sweden. The proportions of AA and ENZ for the periods 2015 to 2018 across the counties are presented in table 1. The fact that the prescription varies over counties is a notable finding as it suggests differences in prescription that may for instance be driven by practice variation. From this table, we can furthermore see that in average across the counties, 24% of the patients were prescribed AA.

### Linked registers
From the NBHW, we link data from the inpatient care register and the pharmaceutical register. All inpatient and outpatient care visits in Sweden and all prescribed drugs are registered in these registers. The inpatient care register contains among others information on all diagnoses (using the International Statistical Classification of Diseases and Related Health Problems 10th Revision, ICD-10), the date of admission and discharge. The pharmaceutical register contains the date of prescribing and dispensing of drugs, and also, the ATC class of the drug.

From SCB, we link data from a census conducted every fifth year over the period 1960–2015; labour statistics based on administrative sources (RAMS) or LISA for the period 1985–2015. RAMS and LISA are large data bases, created by linking a large set of administrative registers using the Swedish person id in the linkage. The linked data contains disposable income, labour income, social insurance payments, capital income, labour market status, year of birth, education, marital status, etc for each individual over the period 1960–2015.

### Confounding
Groups on the different interventions can only be compared in case they are identical on all those variables which simultaneously influence their prescriptions and outcomes. Therefore, we first conducted a qualitative study (Langenskiöld 2021, submitted for publication) with its primary focus of understanding who prescribed which of the drugs, when and for how long.

The response from the interviews where heterogeneous. There is, however, quite good agreement on the following three statements that are relevant for this paper. First, ENZ is preferred for patients with cardiovascular diseases, diabetes, or osteoporosis whereas AA is preferred for patients with poor general health condition and fatigue. Second, doctors want their patients to benefit from as

**Table 1** Proportion prescribed enzalutamide and abiraterone acetate respectively, per county and in total

| County | Enzalutamide | | | | | Abiraterone | | | | |
|---|---|---|---|---|---|---|---|---|---|---|
| | 2015 | 2016 | 2017 | 2018 | TOT | 2015 | 2016 | 2017 | 2018 | TOTAL |
| Blekinge | 0.84 | 0.91 | 0.84 | 0.80 | 0.85 | 0.16 | 0.09 | 0.16 | 0.20 | 0.15 |
| Dalarna | 0.86 | 0.65 | 0.55 | 0.62 | 0.69 | 0.14 | 0.35 | 0.45 | 0.38 | 0.31 |
| Gavleborg | 0.60 | 0.92 | 0.93 | 0.96 | 0.85 | 0.40 | 0.08 | 0.07 | 0.04 | 0.15 |
| Gotland | 0.75 | 0.92 | 1.00 | 1.00 | 0.89 | 0.25 | 0.08 | | | 0.11 |
| Halland | 0.75 | 0.89 | 0.88 | 0.76 | 0.82 | 0.25 | 0.11 | 0.12 | 0.24 | 0.18 |
| Jamtland | 0.47 | 0.81 | 0.77 | 0.67 | 0.70 | 0.53 | 0.19 | 0.23 | 0.33 | 0.30 |
| Jonkopings lan | 0.62 | 0.91 | 0.89 | 0.76 | 0.80 | 0.38 | 0.09 | 0.11 | 0.24 | 0.20 |
| Kalmar | 0.74 | 0.89 | 0.83 | 0.93 | 0.84 | 0.26 | 0.11 | 0.17 | 0.07 | 0.16 |
| Kronoberg | 0.48 | 0.43 | 0.21 | 0.20 | 0.37 | 0.52 | 0.57 | 0.79 | 0.80 | 0.63 |
| Norrbotten | 0.83 | 0.88 | 0.64 | 0.65 | 0.74 | 0.17 | 0.12 | 0.36 | 0.35 | 0.26 |
| Orebro | 0.77 | 0.95 | 0.68 | 0.38 | 0.74 | 0.23 | 0.05 | 0.32 | 0.62 | 0.26 |
| Ostergotlands lan | 0.12 | 0.90 | 0.88 | 0.90 | 0.83 | 0.88 | 0.10 | 0.12 | 0.10 | 0.17 |
| Skane | 0.87 | 0.96 | 0.96 | 0.85 | 0.92 | 0.13 | 0.04 | 0.04 | 0.15 | 0.08 |
| Sodermanland | 0.77 | 0.98 | 1.00 | 0.92 | 0.91 | 0.23 | 0.02 | | 0.08 | 0.09 |
| Stockholm | 0.71 | 0.82 | 0.80 | 0.79 | 0.78 | 0.29 | 0.18 | 0.20 | 0.21 | 0.22 |
| Uppsala | 0.62 | 0.56 | 0.25 | 0.36 | 0.46 | 0.38 | 0.44 | 0.75 | 0.64 | 0.54 |
| Varmland | 0.87 | 0.96 | 0.67 | 0.75 | 0.83 | 0.13 | 0.04 | 0.33 | 0.25 | 0.17 |
| Vasterbotten | 0.63 | 0.70 | 0.62 | 0.76 | 0.67 | 0.37 | 0.30 | 0.38 | 0.24 | 0.33 |
| Vasternorrland | 0.47 | 0.71 | 0.73 | 0.86 | 0.64 | 0.53 | 0.29 | 0.27 | 0.14 | 0.36 |
| Vastmanland | 0.73 | 0.79 | 0.80 | 0.55 | 0.74 | 0.27 | 0.21 | 0.20 | 0.45 | 0.26 |
| Vastra gotalands lan | 0.51 | 0.78 | 0.69 | 0.65 | 0.66 | 0.49 | 0.22 | 0.31 | 0.35 | 0.34 |
| Total | 0.68 | 0.83 | 0.76 | 0.74 | 0.76 | 0.32 | 0.17 | 0.24 | 0.26 | 0.24 |

many treatment options as possible during their lifetime. If it turns out that patients' general health is good and the disease is progressing fast, chemotherapy is the preferred treatment. Instead, if it turns out that patients' general health is poor and the disease is progressing slowly, ENZ or AA is the preferred treatment. Third, doctors monitor the treatment and if it is found that treatments were ineffective or intolerable, the treatment were stopped.

### Covariates

From the inpatient care registers and the pharmaceutical register, we have extensive historical information on patients' health and healthcare consumption. This enables us to create an almost infinite number of covariates. We use health data for the period up to 5 years before the diagnosis, and leave out health data from the date of treatment and onward. That is, all covariates are measured before the first prescription of AA or ENZ and can consequently not be affected by the two treatments.

To exemplify the available data, the medical history from the inpatient care and prescription registry of a randomly chosen patient is presented in figure 1. This patient was diagnosed with a PC in October 2007 (ie, a C61.9 ICD-10 code) and was first treated with ENZ in April 2018 (ie, a L02BB04 ATC code). This patient had no inpatient care visits the years before the cancer diagnosis, and the visits

after the diagnosis is mainly related to the PC diagnose. The patient was hospitalised for more than 1 day in a row at two occasions.

The medical history data are aggregated separately over a 5-year period before the diagnosis and for the period in-between diagnosis and prescription. We derive, among others, the number and length of healthcare visits over the two periods. This is also done separately for patients experiencing metastases, cardiovascular disease, diabetes and fatigue or osteoporosis diagnosis. We also derive the number of specific drugs collected between the cancer diagnose and treatment separately for prescriptions related to these diagnoses.

To gain information on sickness progression, we separately calculate the number of visits the last months before treatment and calculate the average number of visits per quarter between diagnosis and treatment.

Using this strategy, we create 45 continuous covariates where 13 measure the health status before the PC diagnose. We also derive 22 indicator variables, of which 7 measure health status before diagnosis, of whether a patient have had a diagnosis related to metastases, cardiovascular disease, diabetes, fatigue or osteoporosis. See online supplemental table S2 in online supplemental appendix for the included ICD codes.

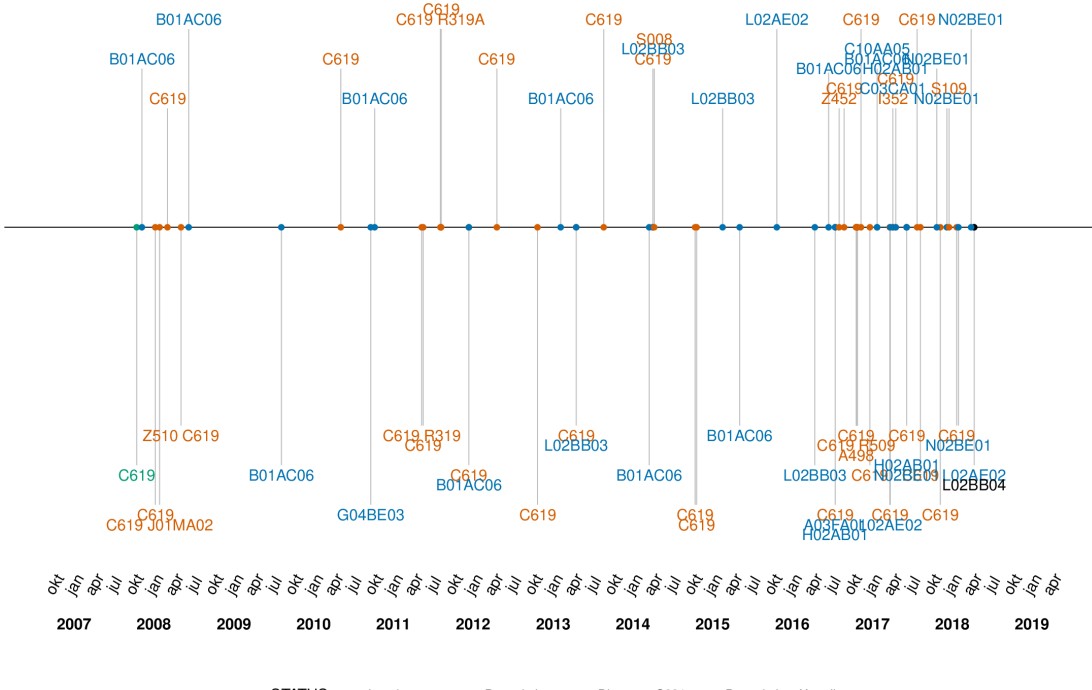

**Figure 1** Description of data on health measures for one randomly chosen enzalutamide-taker.

From the SCB data, we create 94 variables that supposedly will be able to describe the socioeconomic status of the patient 3 years before both the diagnosis and the treatment, with data from 1991 until 2015. For patients with diagnoses before or after these years, data on socioeconomic status is taken from the year closest to the year of diagnosis. This includes information on age, marital status and educational level as well as pensions, income, sick leave and other security benefits for the patient and the household. In the case of missing values, the mean over the three preceding years is used. Educational level is the highest completed education and is classified as less than, equal or more than secondary school.

In addition to the 161 covariates we also add historical county specific mortality in PC. That is, the mean number of men per 100 000 inhabitants who died from PC in 1997–2017. All 162 covariates with descriptions are presented in online supplemental table S3 in Online supplemental

**Table 2** Summary statistics of the AA and ENZ patients for the subset of covariates deemed to be the most important

| Description | ENZ | AA | Diff |
|---|---|---|---|
| Age at treatment | 75.27 (7.85) | 75.29 (7.70) | −0.02 |
| Years to treatment from diagnosis | 6.95 (5.00) | 7.29 (5.31) | 0.34* |
| Less than secondary school education | 0.36 (0.48) | 0.35 (0.48) | 0.01 |
| Secondary school education | 0.39 (0.49) | 0.39 (0.49) | −0.00 |
| More than secondary school education | 0.25 (0.43) | 0.26 (0.44) | −0.01 |
| Living with a partner | 0.66 (0.47) | 0.67 (0.47) | −0.01 |
| Cardiovascular disease before treatment | 0.35 (0.48) | 0.34 (0.47) | 0.01 |
| Atrial fibrillation and flutter before treatment (I48) | 0.16 (0.36) | 0.18 (0.38) | −0.02 |
| Acute myocardial infarction before treatment (I21) | 0.10 (0.30) | 0.07 (0.25) | 0.03*** |
| Other cardiovascular diseases before treatment | 0.27 (0.45) | 0.26 (0.44) | 0.01 |
| Diabetes before treatment | 0.16 (0.37) | 0.12 (0.32) | 0.04*** |
| Osteoporosis before treatment | 0.01 (0.11) | 0.01 (0.10) | 0.00 |
| Secondary malignant neoplasms (metastases) | 0.70 (0.46) | 0.75 (0.43) | 0.05*** |
| Malaise and fatigue | 0.05 (0.22) | 0.04 (0.20) | 0.01 |

SD within parentheses.
*P<0.05, ***p<0.001.
AA, abiraterone acetate; ENZ, enzalutamide.

appendix. Table 2 provides summary statistics of the AA and ENZ patients for the subset of covariates deemed to be the most important. From this table we can see that the two groups in general are very similar, there are for instance no significant differences in average age, educational level or marital status.

## Statistical analysis plan
### Design

A genetic matching approach is used in the design.[9] This is a generalisation of Mahalanobis distance matching where an evolutionary search algorithm is used to maximise the balance of a set of observed covariates together with an estimated propensity score.[10]

Due to computational limitations, we cannot match on all covariates. Thus, we need to restrict the set to the ones deemed the most important. According to the 12 urologists and oncologists interviewed for this project, cardiovascular diseases, fatigue, diabetes, osteoporosis and age at treatment influenced their choice of prescriptions and were thus included. As prevalence of visceral metastases indicated the severity of the disease, and the sequencing of prostate-cancer drugs reflected the optimisation of care according to the same specialists, we also considered presence of visceral metastases and time to prescription. Also socioeconomic variables such as educational level, marital status and quality of care differences across counties (measured by county specific mortality) were included. We also include nine factors that summarise the information from the 130 remaining covariates and an estimated propensity score (see online supplemental table S4 in online supplemental appendix for the factor loading of the set of variables included in the factor analysis).

The factors are derived from an exploratory factor analysis (using the varimax rotation and factor scores derived using the regression method).[11 12] The nine factors account for 42% of the variance in the 130 variables.

The propensity score is estimated using a logit model, and least absolute shrinkage and selection operator (LASSO) regression.[13] All 162 covariates are included, in addition, interactions and second order polynomials of the continuous covariates are included. The LASSO regression has a penalty, or cost, of including too many covariates. This helps us avoid the risk of overparametrising the propensity-score which may lead to bias.[14]

A one-to-one matching with replacement is used in the genetic matching approach, but we exclude observations with a distance above 3 SD for any of the included covariates. This leads to 85 dropped observations and a total of 4 516 matched observations.

The propensity score balance is presented in figure 2. It is clear that the estimated probability to be treated is similar already in the unmatched groups, but the balance is improved in the adjusted sample. The covariate balance in the matched sample is examined through the standardised difference in per cent of the average SD (SB). Further, we examine the variance ratio (VR) between the

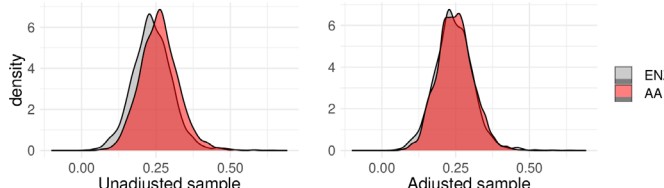

**Figure 2** Propensity score overlap, before and after matching. AA, abiraterone acetate; ENZ, enzalutamide.

two groups for each of the included covariates. In a well conducted randomised experiment, we expect the SB to be no larger than 25% and the VR to be close to one. This is used as a rule of thumb in the quasi-experimental literature.[15] The results are presented in table 3.

## Preanalysis plan

We have one primary outcome, two secondary outcomes and one exploratory outcome. The primary outcome is all-cause mortality (DEAD) and the two secondary outcomes capturing morbidity are PAIN, and SRE. PAIN is an indicator for severe pain, and SRE is an indicator for a skeleton-related events. The reason for including PAIN and SRE as outcomes is that these two morbidities have been seen as common complications of bone metastases.[16 17]

For the inference, we will use Bonferroni correction with a 5% overall level. With one primary and two secondary outcomes, this means that the significance level on the single outcomes will be 1.67% (=100×0.05/3).

DEAD as a primary endpoint is an indicator variable defined as one for patients who are dead of any causes and zero for other patients at the end of each 30 days period after being prescribed AA- or ENZ. Mortality data will be available until the end of June, 2020. Consequently, the first patient administered the treatments can theoretically have up to 70 mortality registrations, but the actual number of registrations will most likely be fewer as they are suffer a deadly disease.

Patients are assumed to suffer severe pain if they receive prescriptions for neuropathic pain, that is, opiates in combination with tramadol and paracetamol (ATC-codes N02AA, N02A×02 and N02BE01). The PAIN indicator is one for periods in which the patient has received such a subscription and zero for the other periods. Prescription data will be available until 31 December 2020. Therefore, up to 76 periods are available for prescription data.

Patients are assumed to suffer skeleton related event if they experience a hospitalisation because of pathologic fracture (ATC codes M485, M495, M844 and M907) or spinal cord compression (G550, G834, G952, G958, G959 and G992).[18] The SRE indicator is one for periods with such hospitalisations and zero for the other periods. As inpatient care data is available until 31 December 2019, at most 64 periods are available.

For each of the periods, we will estimate the above three effects. We will display the Bonferroni adjusted CIs

**Table 3** Standardised difference in percent and variance ratio (VR) before and after matching

|  | SB before | SB after | VR before | VR after |
|---|---|---|---|---|
| Age (years) | 0.28 | 1.10 | 0.96 | 0.90 |
| Diabetes (Y/N) | 12.48 | 0.70 | 0.77 | 0.99 |
| Other CVD (Y/N) | 3.65 | 1.77 | 0.96 | 0.98 |
| Acute myocardial infarction, I21 (Y/N) | 10.99 | 0.00 | 0.72 | 1.00 |
| Atrial fibrillation and flutter, I48 (Y/N) | 5.43 | 0.91 | 1.10 | 0.98 |
| Osteoporosis (Y/N) | 1.79 | 0.00 | 0.85 | 1.00 |
| Fatigue (Y/N) | 4.56 | 0.00 | 0.82 | 1.00 |
| Less than secondary school education | 3.15 | 5.08 | 0.98 | 1.03 |
| Secondary school education | 0.45 | 2.32 | 1.00 | 0.99 |
| Time to treatment (years) | 6.58 | 0.55 | 1.13 | 0.94 |
| Secondary malignant neoplasms (Y/N) | 11.73 | 1.48 | 0.89 | 0.99 |
| Time to treatment squared (years) | 7.75 | 1.26 | 1.23 | 0.88 |
| Age squared (years) | 0.07 | 0.58 | 0.97 | 0.91 |
| Age and diabetes | 12.41 | 0.45 | 0.77 | 1.00 |
| Age and CVD | 3.44 | 1.43 | 0.97 | 0.99 |
| Age and secondary malignant neoplasms | 12.06 | 1.59 | 0.90 | 0.98 |
| Diabetes and CVD | 6.77 | 0.00 | 0.77 | 1.00 |
| County specific mortality | 6.05 | 0.23 | 0.99 | 0.90 |
| Partner (Y/N) | 4.01 | 6.33 | 0.94 | 0.90 |
| Factor 1 | 1.03 | 1.48 | 1.08 | 0.72 |
| Factor 2 | 1.29 | 0.54 | 0.36 | 0.94 |
| Factor 3 | 0.66 | 5.03 | 0.68 | 0.81 |
| Factor 4 | 5.07 | 2.39 | 1.42 | 0.94 |
| Factor 5 | 4.37 | 1.48 | 0.71 | 0.95 |
| Factor 6 | 5.54 | 3.35 | 0.13 | 0.76 |
| Factor 7 | 1.30 | 1.90 | 0.53 | 0.78 |
| Factor 8 | 0.15 | 0.43 | 0.90 | 0.90 |
| Factor 9 | 0.21 | 0.35 | 0.48 | 0.97 |
| Propensity score | 37.03 | 9.10 | 0.76 | 0.83 |

CVD, cardiovascular disease.

(ie, using the 1.67% level for an overall 5% level test) at each time period.

For all outcomes, regression analysis will be used in the estimation of the comparative effectiveness analyses of AA against ENZ. We adjust for all variables displayed in table 2 except for the propensity score. In addition, we allow for heterogeneous effects by also adjusting for the covariates interacted with an indicator for AA. For the inference, we will use the Eicker-Huber-White covariance estimator robust covariance matrix.[19–21]

The problem with analysis of the two morbidity outcomes is that they are only observed in data if the patient is alive. In the analysis above we will, at each evaluation period, remove the dead patients from the analysis. This means that the number of valid observations will be reduced over the evaluation window. If there are no differences in mortality rates between the two drugs

this analysis strategy provides unbiased estimates of the comparative effectiveness on morbidity. However, if there are differences in mortality rates, these analyses are biased as it is more likely that we observe a morbidity for the drug with lower mortality rates. If this is the case, we will need to estimate bounds of potential effects as a sensitivity analysis. We let all patients who die either have the morbidity or not (ie, PAIN=1 and SRE=1 and PAIN=0 and SRE=0). If the mortality is observed to be higher for AA than for ENZ the first case provides the upper bound estimate of the comparative effectiveness of AA against ENZ while the second one provides the lower bound on these two morbidity outcomes and vice versa if the mortality rate is lower for AA.

In addition, we will analyse two explanatory endpoints. The first is prostate-specific mortality and the second is the duration of reversing treatments (COMPLY).

Prostate-specific mortality is measure in 30 days' intervals in the same way as in the main analysis. COMPLY is measured as the duration until the patients is observed to have stopped their treatment. COMPLY is calculated based on prescription data and their average estimated compliance rate.

The problem with analysing prostate-specific mortality is of how to deal with people dying from other causes of death. To this end we repeat the analysis with the two morbidity outcomes, that is, we remove patients who die from other causes and also estimate the upper and lower bound where those who die from other causes are either treated as dying from PC or not dying. All estimates are performed using the regression models explained above. The analysis of DRT will be conducted using Cox regression. Covariates displayed in table 2 will be added in addition to the indicator for AA. We treat mortality as a competing risk conditional on the observed covariates.

## Subgroup analyses

Subanalyses on these four outcomes will be conducted on patients who are believed (1) to be potential high/low responders; (2) to suffer a more aggressive disease and (3) to have had a long or short time until subscription.

Using the results from our qualitative work we define high responder patients in (1) as those who are treated twelve months or more to previous hormone treatment, that is, Luteinizing hormone-releasing hormone (LHRH) agonists or—antagonists, while the low responder is treated shorter. The hormone treatments are leuprolide (ATC L02AE02), goserelin (ATC L02AE03), triptorelin (ATC L02AE04), och histrelin (ATC L02AE05) and degarelix (ATC L02B×02), respectively. This corresponds to 63% of the sample being categorised as high respondents.

We regard patients with visceral metastases in their hospital records (ICD-10 code C78) before they start their AA or ENZ treatment in (2) as having have more aggressive disease.[22 23] About 7% of the patients are defined as suffering from a more aggressive disease.

All patients with a time between diagnosis of PC and prescription for AA or ENZ treatment shorter than the median waiting time for both drugs are defined as patients with short time until subscription (3) while the complement is patients with long time.

## Sensitivity analyses

While the matching design yields balanced observed covariates, this is an observational study with the usual limitations in this context. In particular, one cannot discard the possibility that unobserved confounders are not balanced. However, Rosenbaum's sensitivity analysis for hidden bias will be used.[24] McNemar's test will be used when calculating ranges of p values of the Intent to treat effect for different values of the sensitivity parameter.[25]

In addition, we will add data from the national prostate cancer register (NPCR). These data contain more detailed information on patient's health with regard

to the PC but do not have full coverage. The data are ordered and we will have access to it at the same time as we have the mortality data. These data allow us to basically test the validity of the design by estimating the effects on potential pretreatment confounders. If there is a statically significant effect on these covariates this suggests that available data from population register are not sufficient to control for confounding bias.

For the test, we use three covariates, that during discussions with specialists, are judged to be important confounders: PSA levels (D_SPSA), Gleason score (D_GleasSa) and metastases (D_Mstad) at the time of PC diagnoses. With three premeasured covariates, as in the main analysis, we adjust the significance level for the individual tests using Bonferroni correction based on a 5% overall level. This means that the level for testing on each single outcome is 1.67%. For all three covariates, regression analysis will be used in the estimation using the same regression designs as for the main analysis. NPCR also allows us to do more exploratory subgroup analyses. As one of the drug company we consulted believed that the initial treatment may influence the effectiveness of their drug, we will separately evaluate patients who receive conservative therapy (wait and see), radical prostatectomy, and radiation therapy as their primary treatment.

Generalisability of the results to other populations may not necessarily be granted, for instance, if future treated populations differ greatly in characteristics, which modify the effect of the treatment.

## DISCUSSION

This protocol has specified a design and a detailed preanalysis plan for a comparative effectiveness analysis using administrative registers. As in an Randomized Controled Trial (RCT), the focus is on the design of the assignment mechanism before observing the outcome. The matching design creates an analysis sample of the two treatment groups that are as comparable on observed covariates as in an RCT. The approach lend objectivity to our results. Contrary to RCT, we can, however, not assure the two treatment groups are comparable on unobservable covariates. For that reason, sensitivity tests for the maintained assumption of unconfoundedness are presented in the protocol.

## Public and private involvement

Clinicians (oncologists and urologist) were involved in previous works for this protocol development. The clinicians were interviewed to understand factors that are important while prescribing AA and ENZ.

## Ethics and dissemination

Ethical approved by the Ethical committee in Uppsala (ref. Dnr2017/482). Results will be published in a peer-reviewed journal and distributed to healthcare providers, healthcare policy-makers and potentially patients by way

of printed and electronic materials as well as oral presentation at regional meetings.

**Acknowledgements** We would like to acknowledge the valuable contribution of the urologists and oncologists who participated in our qualitative work (R. Blom, G. Ahlgren, Å. Jellvert, M. Anden, K. Holmsten, S-O. Andersson, L. Nygren, L. Åström, M. Hellström, C. Thellenberg, J. Yachnin, J. Sandzen). They helped us understand what factors govern the prescription of AA, ENZ, that is, factors which needs to balance in any groups compared in a comparative effectiveness evaluation. Also, we would like to thank the groups of experts with whom we intervened in order to improve our protocol, that is, either the balance across the groups or the endpoints to focus on (A. Bill-Azelsson, D. Robinson, R. Henriksson, and representatives from Astellas Pharma and Janssen, that is, Kerstin Magnusson, Johanna Svensson and Ingrid Tredal and Julia Rockberg, respectively. Finally, we thank Sven-Åke Lööf for sharing his great understanding of the field.

**Contributors** PJoh and SL conceived the main conceptual ideas to replicate a randomised study ex-post using Swedish population registries. SL lead and planned the project, and carried out the preparatory qualitative work. PJoh together with PJon worked out the technical details in discussion with SL, and performed the analytical computations. All authors discussed the results and contributed to the final manuscript although PJoh wrote the main part.

**Funding** This work was supported by the Dental and Benefit Agency (diary number 02823/2017).

**Competing interests** None declared.

**Patient consent for publication** Not applicable.

**Provenance and peer review** Not commissioned; externally peer reviewed.

**ORCID iDs**
Per Johansson http://orcid.org/0000-0001-6140-9123
Sophie Langenskiöld http://orcid.org/0000-0003-3888-2910

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
