## [Reviewer comments · BMJ Open]

ARTICLE DETAILS

TITLE (PROVISIONAL)	A study protocol for a comparative effectiveness evaluation of Abiraterone Acetate against Enzalutamide: A longitudinal study based on Swedish administrative registers
AUTHORS	Johansson, Per; Jonéus, Paulina; Langenskiöld, Sophie

VERSION 1 – REVIEW

REVIEWER	Chung, Doo Inha University School of Medicine, Department of Urology
REVIEW RETURNED	01-Jul-2021

GENERAL COMMENTS	This study is a research protocol that will be conducted using large-scale health insurance data. It looks like a well-planned research protocol. However, as the authors said, I think it would be better if the results were attached together and compiled into one paper. I recommend that you add the results by performing statistical analysis. Thank you.
---

REVIEWER	Shore, Neal Carolina Urologic Research Center
REVIEW RETURNED	04-Jul-2021

GENERAL COMMENTS	The undertaking for this detailed, retrospective, comparative analysis of 2 distinctive AR antagonists is relevant. The authors have ambitiously embarked upon a very detailed analysis of numerous variables. My overarching criticism is that the manuscript is very difficult to read and requires substantial rewording.
--

REVIEWER	Maughan, Benjamin University of Utah
REVIEW RETURNED	18-Aug-2021

GENERAL COMMENTS	This is well written and appropriate for publication. The methods are adequately described. The comparative effectiveness of these NHTs is still an important topic to the field. Some real world data (which this provides) is lacking. The only suggestion is in the introduction the authors use the phrase novel antiandrogenic (NAM). This is a non-standard
--

abbreviation. I suggest they modify to use novel hormone therapy (NHT) which is more standard in the field.

VERSION 1 – AUTHOR RESPONSE

	Dr. Doo Chung’s comments:	Our comments:
1	This study is a research protocol that will be conducted using large-scale health insurance data. It looks like a well-planned research protocol.	We are grateful that Dr. Chung understand the effort we have invested in planning the way to design and analyze our observational study.
2	However, as the authors said, I think it would be better if the results were attached together and compiled into one paper. I recommend that you add the results by performing statistical analysis.	We agree that we were not clear in the previous manuscript why it is important to publish protocols also for observational study, i.e., to avoid “p-hacking”. We add this justification under the heading “Discussion”, which is found both in the abstract and in the running text.
	Dr. Neal Shore’s comments:	Our comments:
1	The undertaking for this detailed, retrospective, comparative analysis of 2 distinctive AR antagonists is relevant	We are grateful that Dr. Shore acknowledge our comparative effectiveness analyses as relevant.
2	The authors have ambitiously embarked upon a very detailed analysis of numerous variables.	We believe that this is an important remark by Dr Shore and which emphasize the strength of our comparative effectiveness evaluation, i.e., that we have compared numerous variables across the groups of patients on abiraterone acetate and enzalutamide and found no differences. Consequently, if we find a difference in the analyses, we believe it is caused by the treatments and no other differences.
3	My overarching criticism is that the manuscript is very difficult to read and requires substantial rewording.	We have revised and restructured the paper to make the protocol more readable. A colleague reviewed the manuscript and helped with the wording.
	Dr. Benjamin Maughan’s comments:	Our comments:
1	This is well written and appropriate for publication.	We are glad to hear that Dr. Maughan approve to the way we write, and the appropriateness of publishing the paper.
2	The methods are adequately described.	We are happy to learn that Dr. Maughan agree in the way we describe our methods.
	The comparative effectiveness of these NHTs is still an important topic to the field. Some real world data (which this provides) is lacking.	We thank Dr. Maughan for emphasizing that the specific comparative effectiveness evaluation that this protocol details, will fill a gap of knowledge.
3	The only suggestion is in the introduction the authors use the phrase novel antiandrogenic (NAM). This is a non-standard abbreviation. I suggest they	We are grateful for the comment and we have changed accordingly in the present version of the manuscript.

	modify to use novel hormone therapy (NHT) which is more standard in the field.	
--	--	--